# Early Motor Cortex Connectivity and Neuronal Reactivity in Intracerebral Hemorrhage: A Continuous-Wave Functional Near-Infrared Spectroscopy Study

**DOI:** 10.3390/s25206377

**Published:** 2025-10-15

**Authors:** Nitin Kumar, Geetha Charan Duba, Nabeela Khan, Chetan Kashinkunti, Ashfaq Shuaib, Brian Buck, Mahesh Pundlik Kate

**Affiliations:** 1Department of Biological Sciences, Faculty of Science, University of Alberta, Edmonton, AB T6G 2R3, Canada; nkumar7@ualberta.ca; 2Department of Electrical and Computer Engineering, Faculty of Engineering, University of Alberta, Edmonton, AB T6G 2R3, Canada; geethach@ualberta.ca; 3Division of General Internal Medicine, Department of Medicine, University of Calgary, Calgary, AB T2N1N4, Canada; nabeela.khan@ucalgary.ca; 4Division of Neurology, Department of Medicine, University of Alberta, Edmonton, AB T6G 2R3, Canada; kashinku@ualberta.ca (C.K.); ashfaq.shuaib@ualberta.ca (A.S.);

**Keywords:** acute stroke, intracerebral hemorrhage (ICH), resting-state functional connectivity (RSFC), functional near-infrared spectroscopy (fNIRS)

## Abstract

**Highlights:**

**What are the main findings?**
A seed-based resting-state functional connectivity (RSFC) and motor-paradigm-based oxygenation change assessment using continuous-wave functional near-infrared spectroscopy is feasible in patients with acute intracerebral hemorrhage (ICH).In patients with left hemispheric ICH, RSFC may be increased between the affected primary motor cortex (priMC) and the affected premotor cortex (preMC). In contrast, in right hemispheric ICH, RSFC may be decreased between the unaffected priMC and the affected somatosensory cortex.In patients with right hemispheric ICH with left hand finger tapping, there may be increased oxygenation over the unaffected preMC.

**What is the implication of the main finding?**
Motor cortex reorganization in patients with acute ICH is based on the side of the stroke.Left preMC connectivity and activity may be affected early in patients with ICH, which may serve as a target for neuromodulation devices.

**Abstract:**

Insights into motor cortex remodeling may enable the development of more effective rehabilitation strategies during the acute phase. We aim to assess the affected and unaffected motor/premotor/somatosensory cortex resting state functional connectivity (RSFC) and reactivity with continuous wave functional near-infrared spectroscopy (cw-fNIRS) in patients with ICH compared to age, sex, and comorbidity-matched subjects. We enrolled patients with acute–subacute hemispheric ICH (*n* = 37; two were excluded due to artifacts) and grouped them according to the side (right and left) of the stroke. Matched participants or patients with recent transient ischemic attack were enrolled as control subjects for the study (*n* = 44; five were excluded due to artifacts). RSFC was assessed in both affected and unaffected hemispheres by group-level seed-based (primary motor cortex, priMC) correlation analysis. FT-associated relative oxyhemoglobin (ΔHbO) changes were analyzed in affected and unaffected hemispheres with generalized linear model regression. In left hemispheric ICH, the resting state coherence between the affected priMC and the affected premotor cortex (preMC) increased (*β* = 0.83, 95% CI = 0.19, 1.47, *p* = 0.01). In contrast, in right hemispheric ICH, the coherence between the unaffected priMC and the affected preMC decreased (*β* = −0.6, 95% CI = −1.12, −0.09, *p* = 0.02). In the left hemispheric ICH, the left-hand FT was associated with increased ΔHbO over the affected preMC (*β* = 0.01, 95% CI = 0.003, 0.02, *p* = 0.01). In contrast, in right hemispheric ICH, the left-hand FT was associated with increased ΔHbO over the unaffected preMC (*β* = 0.02, 95% CI = 0.006, 0.04, *p* = 0.01). Left hemispheric preMC may be involved in motor cortex reorganization in acute ICH in either hemisphere. Further studies may be required to assess longitudinal changes in motor cortex reorganization to inform acute motor rehabilitation.

## 1. Introduction

Intracerebral hemorrhage (ICH) accounts for 5–40% of stroke admissions worldwide and is a leading cause of morbidity and mortality. Patients with ICH lead to debilitating motor deficits resulting from the direct or indirect disruption of the motor cortex [1]. Acute ICH management is rapidly advancing, with the implementation of the acute ICH care bundle and the surgical evacuation of the hematoma. However, recovery post-ICH is variable [2]. The severity of neurological deficits is currently assessed clinically using the National Institute of Health Stroke Scale (NIHSS) and with imaging (utilizing computed tomography or magnetic resonance imaging (MRI)). The functional outcomes are assessed using the modified Rankin scale (mRS) or the Glasgow Outcome Scale [3]. These measures have limitations as patients with similar-sized ICH or similar severity of neurological deficits have a variable prognosis. These tools primarily rely on gross clinical observations and fail to capture the underlying neural mechanisms driving recovery [4]. There is a felt need for adjunct quantitative assessment tools that can better predict functional recovery and inform tailored rehabilitation strategies.

Neurophysiological tools, including electroencephalography (EEG), near-infrared spectroscopy (NIRS), and functional MRI (fMRI), may help in understanding complex dynamics of brain network reorganization during recovery. These modalities have higher time and spatial resolution. Acute EEG can predict post-stroke functional disability with a moderate correlation with short-term NIHSS and mRS [5].

However, gel-based EEG applications may not be feasible in acute situations. fMRI, likewise, though a highly reproducible modality, is also resource-intensive and requires the availability of research MRI and stability of patients for testing. Functional near-infrared spectroscopy (fNIRS) is developing as a preferred modality in the acute stroke scenario due to its ease of use [6]. fNIRS is a non-invasive, portable, and cost-effective technique that enables the monitoring of brain activity through the measurement of hemodynamic responses. Its advantages make it particularly suitable for longitudinal studies and bedside applications, bridging the gap between research and clinical practice [7]. Recent studies have successfully employed fNIRS to assess motor cortex activity and functional connectivity in stroke patients, providing valuable insights into hemodynamic alterations associated with recovery [8].

Resting-state functional connectivity (RSFC) and paradigm-based assessments (motor or non-motor) are useful tools in stroke research. RSFC provides insights into the temporal coherence of neural activity across distinct brain regions during rest [9]. RSFC demonstrates blood-oxygen-level-dependent temporal coherence of signals between spatially remote brain regions [10]. Studies have revealed disruptions in brain network dynamics following strokes, particularly within the motor cortex [11]. These disruptions often manifest as altered connectivity within motor regions or between hemispheres, correlating closely with clinical motor impairments. Despite its utility, RSFC studies using fMRI have several limitations, including high cost, limited accessibility, and constraints on real-time applications in clinical settings [12].

Response to the Motor task paradigm-based assessment can help understand cortical reorganization in the phases of acute, subacute, and chronic stroke. This may enable tracking the trajectory of recovery after a stroke.

We aimed to assess RSFC in both affected and unaffected hemispheres by group-level seed-based (primary motor cortex, priMC) correlation analysis in patients with ICH. We also aimed to measure finger-tapping-associated relative Oxyhemoglobin changes (ΔHbO) in affected and unaffected hemispheres in patients with ICH.

We hypothesize that ICH leads to both intra- and inter-hemispheric disruptions in motor network connectivity, resulting in altered hemodynamic responses that correlate with clinical motor impairments. Specifically, we anticipate that task-induced hemodynamic changes will offer additional markers of motor network disruption, providing a nuanced understanding of how ICH affects motor control.

## 2. Materials and Methods

### 2.1. Study Design and Patient Population

The study enrolled adult patients diagnosed with acute ICH at the tertiary care center. Participants included individuals with both left- and right-hemispheric strokes. Eligibility criteria required patients to be 18 years or older with a confirmed ICH diagnosis, while exclusion criteria included conditions that could interfere with fNIRS measurements, such as asymptomatic severe cervical or internal carotid stenosis, ongoing hemodynamic instability, use of antiseizure medication, preexisting hepatic, respiratory, or renal disease, and symptomatic hypothyroidism. A control group, matched by age, sex, and comorbidities, was recruited to provide baseline comparisons. Controls had no history of stroke or presented only with a transient ischemic attack (TIA) without lasting neurological impairment. Ethics approval was obtained from the Institutional Review Board, and informed consent was secured from all participants or through their legally authorized representatives.

### 2.2. Study Procedures

Stroke severity was evaluated using the NIHSS, ranging from 0 (no symptoms) to 42 (death), at baseline (<24 h symptom onset), 2 h, 24 h, 7 days, and 90 days. Functional disability was measured at 90 days using the mRS, where 0 indicates no disability and 6 represents death.

### 2.3. fNIRS Data Acquisition

Data were collected using the NIRSport2 (NIRx Inc., Berlin, Germany), a mobile fNIRS system, at a single time point. The device operated at 760 nm and 850 nm wavelengths, allowing for the quantification of ΔHbO and deoxygenated hemoglobin (HbR) as markers for cerebral oxygenation and blood flow in the tissue underlying the sensors.

For each participant, fNIRS data were collected from 16 channels over the premotor, primary motor, and somatosensory cortices. The eight optodes (light-emitting sources) and eight detectors (Avalanche Photodiodes) were placed on the participant’s scalp in accordance with the international 10-10 EEG electrode placement system (Figure 1 and Appendix A). The distance between each optode and detector was standardized at 35 mm, and the sampling rate was set at 10.2 Hz. The collected data were recorded using the NIRSite version 2.0 software (NIRx Inc., Berlin, Germany). Real-time signal assessment was performed using the Aurora fNIRS software, which allowed for continuous monitoring of each optode’s signal quality. The signal-to-noise ratio (SNR) was calculated in real-time using the coefficient of variation, and only channels with an SNR below 15% were retained for further analysis. A 3 min resting-state fNIRS recording was completed, followed by a motor paradigm that included finger-tapping exercise. The motor paradigm was implemented using PsychoPy Builder (v2022.2.5, University of Nottingham, Nottingham, UK) and was conducted in the following sequence:


*The finger tapping task required participants to sequentially tap each finger to their thumb at a rate of one tap per second. The task alternated between hands, with each hand completing 10 repetitions, resulting in a total of 5 alternations. The entire task lasted 3 min, with a 3 s pause between stimuli to allow brief rest intervals.*


This motor protocol was administered to all participant groups, including those with ICH and controls. Data analysis was conducted using Satori fNIRS software version 2.0 (NIRx Inc., Berlin, Germany).

### 2.4. fNIRS Data Processing

The raw light intensities from the fNIRS recordings were converted into relative changes in ΔHbO and HbR concentrations, leveraging the modified Beer–Lambert law. This transformation from optical density (OD) to concentration chromophores (CC) enabled the quantification of hemodynamic responses within specified brain regions. Two primary areas associated with motor function—the primary motor cortex and premotor cortex—were identified as key regions of interest. Channels aligned with these regions were selected for further analysis, guided by the signal quality and spatial coverage of the optode-detector layout.

All raw data were saved in the standardized .snirf format and systematically organized into structured directories specific to each patient and control subject as per the guidelines for fNIRS analyses [13]. A consistent file naming convention was applied to ensure uniformity across datasets, incorporating metadata on acquisition time, patient ID, and task type (e.g., resting state or finger tapping). Data processing involved filtering, noise reduction, and artifact correction (Appendix A). As shown in Figure 1, distinct pre-processing pipelines were established for the resting-state and motor-task data to accommodate the differing analytical requirements of each task.

### 2.5. Workflow Manager Data Pipeline

Scalp Coupling Index (SCI): An SCI threshold of 75% ensured reliable fNIRS recordings, with channels below this threshold excluded to reduce noise from poor signal contact with the scalp [14]. This threshold was critical for maintaining data quality, particularly for patients with ICH who may have scalp conditions that affect sensor coupling.

Pre-Whitening: A 5 s buffer was removed around each event to mitigate baseline effects from pre-task variations, optimizing the accuracy of fNIRS signal comparisons across tasks. By accounting for baseline shifts, this step also minimized potential signal drifts common in fNIRS data from acute ICH patients.

Spike Removal: Motion artifacts were minimized using a spike-removal algorithm, with settings customized per task [15].

Monotonic Interpolation and Temporal Derivative Distribution Repair (TDDR): Monotonic interpolation was used to smooth the data, followed by TDDR, which corrected for motion artifacts that may otherwise distort hemodynamic measurements [16]. This method was particularly relevant for motor tasks where head and body movement could influence sensor readings.

Temporal Filtering: A Butterworth filter was applied to reduce noise with a low-pass cutoff set at 0.15 Hz and a high-pass at 0.01 Hz for motor tasks (and at 0.10 Hz for resting data). These parameters were optimized to maintain frequency ranges that reflect meaningful hemodynamic changes without the interference of high-frequency noise.

Global Component Regression (GCR) and Correlation-Based Signal Improvement (CBSI): GCR was employed to filter global noise from the fNIRS data, thereby addressing artifacts arising from systemic physiological changes, such as heart rate and respiration. Additionally, CBSI corrected for head movement artifacts, essential for isolating the hemodynamic activity related to motor cortex responses with minimal interference from external factors [17].

Short Channel Regression (SCR): SCR was applied to remove superficial physiological noise originating from the scalp and extracerebral tissues. This enhances the specificity of cortical signals by regressing out non-neuronal activity captured by short-distance optodes. The “highest-correlated” setting was used in SCR, selecting the short channel most temporally aligned with each long channel, ensuring optimal noise reduction and preserving task-related hemodynamic responses with minimal contamination from surface-level artifacts.

Normalization: Z-transform and percent signal change (PSC) normalizations were implemented to standardize data and adjust for motion artifacts, allowing for reliable inter-participant and inter-hemispheric comparisons [18]. These normalizations ensure data consistency, critical for analyzing the effects of ICH on bilateral motor cortex functionality.

Patients with at least one symmetric sensory channel were retained for further analysis. This ensured that fNIRS data accurately captured bilateral oxygenation and hemodynamic responses. Patients with data that did not meet these criteria, due to poor signal quality or asymmetry between the sensory channels, were excluded from the data analysis. This approach ensured data reliability and validity when comparing interhemispheric hemodynamics (Appendix A).

### 2.6. fNIRS Analyses

fNIRS analysis was conducted for resting-state functional connectivity (RSFC) and motor paradigm.

#### 2.6.1. Resting-State Analyses

RSFC was assessed via seed-based correlation; priMC served as the seed region, with the analysis focusing on the connectivity between the affected and unaffected hemispheres. Seed-based correlation is particularly suited for examining connectivity patterns between a predefined region of interest (the “seed”) and other brain regions, enabling targeted analysis of synchronized activity across brain networks.

#### 2.6.2. Motor Paradigm Analyses

A general linear model (GLM) analyzed motor task data and stroke laterality (left or right hemisphere affected) served as a predictor in the model. For left-hemispheric stroke patients, the contrast was set to Left Hand > Baseline (1 > Baseline), and for right-hemispheric stroke patients, it was set to Right Hand > Baseline (2 > Baseline). Beta (*β*) coefficients from GLM results quantified motor task-related brain activation.

The GLM provides an established analytical framework for quantifying brain activity in response to task conditions by modeling the relationship between experimental predictors and observed neural responses, accounting for both fixed and random effects to detect significant hemodynamic responses while controlling for inter-subject variability [19].

False discovery rate (FDR) correction was applied under the GLM analysis to control for multiple comparisons across the 26 fNIRS channels. The Benjamini and Yekutieli method was selected with an alpha level of 0.05, ensuring a balanced approach to minimizing Type I error while maintaining sensitivity to true effects.

#### 2.6.3. Outcome Measures

The primary outcome measure is RSFC and relative changes in ΔHbO during finger tapping tasks in patients with ICH. RSFC evaluates disruptions in coherence between motor-related brain regions in the bilateral hemispheres. A reduction in ΔHbO concentrations in affected motor regions indicates impaired neural reactivity, whereas increased values in other areas may suggest compensatory activity.

### 2.7. Statistical Analyses

We had two groups of participants: ICH and control participants. Categorical or nominal variables were expressed as proportions and compared with the chi-square test for (female sex, comorbidities). Continuous variables were expressed as means or medians and compared using Student’s *t*-test or the Kruskal–Wallis test (for age and ICH volumes).

RSFC was assessed in individual affected and unaffected hemispheres by group-level seed-based correlation analysis (seed placed on priMC). Additionally, linear regression analysis was performed with controls and stroke patients by selecting a motor channel on the affected and unaffected hemispheres separately. Finger tapping-associated ΔHbO changes were analyzed in affected and unaffected hemispheres with general linear model (GLM) regression. Correlations between left and right motor cortices and event-related averages were calculated for hemodynamic responses in both resting and motor-task states. We assessed functional outcome at 90 days with the mRS. Predictors of functional outcome (age (years)), NIHSS at the time of fNIRS, hematoma volume (mL), and left premotor cortex oxygenation changes with finger tapping (ΔHbO) were assessed with ordinal logistic regression. Significance was determined with a threshold of *p* < 0.05. All analyses were conducted using STATA 18.0 BE (StataCorp LLC, College Station, TX, USA).

## 3. Results

### 3.1. Patient Characteristics

In this cross-sectional study, we enrolled 37 patients with ICH and 44 control/TIA participants. Seven (Two ICH and five control/TIA patients) were excluded from the study due to poor fNIRS signal quality during the analysis. The 35 patients with ICH included in the analyses were enrolled at a median (IQR) of 42.1 (22.6, 88.3) hours after symptom onset, with a median NIHSS of 10 (5, 17); 42.8% were female, and 19 (54.3%) were right hemispheric in location. Table 1 describes the comorbidity burden between the patients with ICH and control/TIA participants. Dyslipidemia was more common in control/TIA participants, 30 of 39 (76.9%), and the small vessel disease burden was higher in patients with ICH, 24 of 35 (68.6%).

### 3.2. Resting State Functional Connectivity Analysis

RSFC analysis in ICH was performed using seed-based correlations placed on the right or left hemisphere priMC compared to control/TIA participants.

When the seed was placed on the left hemisphere priMC in patients with left hemispheric ICH, there was an increased coherence with the affected premotor cortex (preMC, FC4-FC2, *β* = 0.83, 95% CI = 0.19, 1.47, *p* = 0.01) and a decreased coherence with the affected primary somatosensory cortex (C3-C5, *β* = −0.76, 95% CI = −1.4, −0.13, *p* = 0.02), compared to control/TIA (Figure 2).

No coherence differences were observed in the remaining affected or unaffected cerebral regions between left hemispheric ICH and controls/TIA participants.

In patients with right hemispheric ICH, when the seed was placed on the left hemisphere priMC there was a decreased coherence with affected primary somatosensory cortex (CP2-FC2, *β* = −0.71, 95% CI = −1.32, −0.09, *p* = 0.02) and the unaffected preMC (FC3-FC1, *β* = −0.6, 95% CI = −1.12, −0.09, *p* = 0.02). No coherence differences were observed in the remaining affected or unaffected cerebral regions between right hemispheric ICH and controls/TIA participants.

### 3.3. Motor Paradigm (Finger-Tapping) Analyses

Motor paradigm (Finger tapping) analyses were done with generalized linear regression for each channel. The control subjects and TIA participants served as a comparator group (Figure 3 and Figure 4).

In left hemispheric ICH, with left hand finger-tapping, the ΔHbO was higher in the left (affected) somatosensory cortex (CP3-C3, *β* = 0.02, 95% CI = 0.005, 0.03, *p* = 0.008) and in the left (affected) preMC (C3-FC3, *β* = 0.01, 95% CI = 0.003, 0.02, *p* = 0.01).

In left hemispheric ICH, with left hand finger-tapping, the HbR change was lower in the left (affected) somatosensory cortex (CP3-C3, *β* = −0.02, 95% CI = −0.03, −0.005, *p* = 0.01) and the left (affected) preMC (C3-FC3, *β* = −0.01, 95% CI = −0.02, −0.002, *p* = 0.02).

In right hemispheric ICH, with left hand finger-tapping, the ΔHbO change was higher in the left (unaffected) preMC (C3-FC3, *β* = 0.02, 95% CI = 0.006, 0.04, *p* = 0.01). The HbR change was lower in the left (unaffected) preMC (C3-FC3, *β* = −0.07, 95% CI = −0.044, −0.005, *p* = 0.01).

The total hemoglobin (Hb) change was lower in the left (unaffected) somatosensory cortex (C1-C3, *β* = 0.02, 95% CI = −0.13, −0.007, *p* = 0.03) and higher in the right (affected) preMC (FC4-FC2, *β* = 0.08, 95% CI = 0.01, 0.15, *p* = 0.02).

No difference was observed in ΔHbO over either hemisphere in other channels with either right-hand or left-hand finger tapping compared to control/TIA participants. No difference was observed in HbR over either hemisphere in other channels with either right-hand or left-hand finger tapping compared to control/TIA participants. No difference was observed in total Hb over either hemisphere in other channels with either right-hand or left-hand finger tapping compared to control/TIA participants.

Figure 4 explains group-level comparisons of control, left, and right ICH stroke. In map 1 (Left Hand > Baseline in left priMC), control differed significantly from both left-hemispheric ICH (*p* < 0.01, *d* = 0.86, large) and right-hemispheric ICH (*p* < 0.01, *d* = 0.87, large), while no significant difference was observed between left- and right-hemispheric ICH (*p* = 0.84, *d* = 0.07, negligible).

In map 2 (Right Hand > Baseline in right priMC), controls did not differ significantly from left-hemispheric ICH (*p* = 0.64, *d* = 0.12, negligible) and right-hemispheric ICH (*p* = 0.18, *d* = −0.35, small), while left- and right-hemispheric ICH did not differ significantly as well (*p* = 0.07, *d* = −0.65, medium).

In map 3 (Left Hand > Baseline in right priMC), controls did not differ significantly from left-hemispheric ICH (*p* = 0.51, *d* = 0.22, small) or right-hemispheric ICH (*p* = 0.06, *d* = 0.48, small). Similarly, no significant difference was observed between left- and right-hemispheric ICH (*p* = 0.52, *d* = 0.25, small).

In map 4 (Right Hand > Baseline in left priMC), there was no significant difference observed between controls and left-hemispheric ICH (*p* = 0.64, *d* = 0.12, negligible) or right-hemispheric ICH (*p* = 0.18, *d* = −0.35, small). Similarly, there was no significant difference between left- and right-hemispheric ICH (*p* = 0.07, *d* = −0.65, medium).

### 3.4. Motor Paradigm (Handgrip) Analysis

Motor paradigm (Handgrip) analyses were done with generalized linear regression for each channel. The control subjects and TIA participants served as a comparator group. No differences were observed in ΔHbO, HbR, and total Hb over either hemisphere in channels with either right-hand or left-hand grip compared to control/TIA participants.

### 3.5. Functional Neurological Outcome

The median modified Rankin scale (mRS) at 90 days was 4 (2, 5), and 6 (17.1%) patients died. Age (odds ratio 0.89, 95% CI 0.81–0.97, *p* = 0.01) and NIHSS at the time of fNIRS recording (OR 0.83, 95% CI 0.71–0.98, 0.02) were predictors of mRS at 90 days. However, left PreMC (FC3-FC1, FC3-FC5) was not a predictor of mRS at 90 days.

## 4. Discussion

This study demonstrated that RSFC over the priMC is affected early (within 48–72 h) in patients with ICH. Furthermore, in patients with left hemispheric ICH, there is increased coherence on seed-based analysis with the affected preMC compared to controls/TIA participants. In the motor paradigm assessment, patients with left hemispheric ICH exhibit increased oxygenation, a surrogate for neuronal activity, over the affected preMC and affected somatosensory cortex. In contrast, those with right hemispheric ICH show increased oxygenation over the unaffected preMC alone.

Increased coherence may reflect either a maladaptive hyperconnectivity, a compensatory phenomenon, or a combination of the two [20]. In fMRI studies, it is observed that cortical reorganization occurs either in the affected hemisphere (increased coherence in the supplementary motor area or anterior cingulate gyrus) or in the unaffected hemisphere (supplementary motor area or priMC) [21]. However, over the subacute and chronic phases, there is pruning of this cortical reorganization to a small, well-defined region either in the affected or unaffected hemisphere [22,23,24].

Patients with ICH exhibited reduced coherence between the affected and unaffected hemispheres, highlighting impaired inter-hemispheric communication. Specifically, the negative coefficient observed for C3-C5 (sensory/AT) in left-stroke patients indicates weakened coherence between sensory and motor areas, which may contribute to motor impairments. Conversely, increased connectivity in FC4-FC2 (preMC) indicates compensatory mechanisms, suggesting an adaptive response to motor deficits due to the increased coherence observed between the priMC and preMC.

The motor paradigm results further support these findings, as fNIRS revealed differences in hemodynamic responses between patients with ICH and control subjects. Hemodynamic activity in the affected motor regions was notably reduced in stroke patients, indicating impaired cortical activation. Specifically, left-hemispheric stroke patients showed diminished motor cortex activation, with lower oxyhemoglobin levels during movement tasks. The C4-FC4 (Motor) channel exhibited a reduction in both oxy- and deoxyhemoglobin concentrations, emphasizing impaired neural activation (Figure 3). In contrast, right-hemispheric ICH patients exhibited increased activity, particularly in the C2-FC2 motor and premotor regions, suggesting affected side reorganization of neurons to mitigate the motor deficits caused by ICH (Figure 3).

In previous resting state fMRI (rsfMRI) studies, dysconnectivity has been observed in the subacute and chronic RSFC in acute ICH. In a cross-sectional study, reduced coherence in the default mode network (DMN) and orbitofrontal cortex was observed in unresponsive patients using seed-based functional connectivity. This suggests early disruption in the frontal network in the acute phase [20]. In addition, a longitudinal study compared ICH patients with matched controls using RSFC. The ICH group experienced reduced coherence between DMN and sensorimotor networks in 1 month after stroke. Over time, the ICH group experienced a minimal increase in coherence in 3 months; however, the coherence continued to increase over 12 months. These findings suggest that there are longitudinal changes from subacute to chronic phases of RSFC in fMRI studies. Comparable changes have also been observed in previous fNIRS studies, specifically in a cohort of ICH patients, where RSFC in the subacute phase was found to decrease coherence in functional connectivity between the dorsolateral prefrontal cortex (DLPFC) and the bilateral primary motor cortex. These results demonstrate that RSFC changes by fNIRS are similar to those observed in rsfMRI, particularly in frontal-motor circuits.

Building on RSFC changes, studies have also examined motor network reorganization in patients with ICH. An fMRI study revealed patterns of motor reorganization in patients with acute ICH, characterized by disrupted temporal variability in regional brain activity that evolves across recovery stages. During the acute phase, patients exhibit reduced temporal activation in the affected precentral gyrus (preCG). As recovery progresses to the subacute stage, activation increases in the preCG, preMC, and sensory regions. Notably, the degree of increased preCG activation from the acute to the subacute stage correlates with long-term motor improvement. This aligns with an fNIRS study, which tracked sensorimotor reorganization in acute ICH patients. Like fMRI, fNIRS revealed early bilateral activation during paretic-arm movements, followed by progressive lateralization to the affected hemisphere. Both studies highlight an acute-phase loss of lateralization due to the unaffected hemisphere’s compensation, and a subsequent subacute restoration of the affected hemisphere’s dominance during recovery.

In this study, cw-NIRS was used as a non-invasive neuroimaging technique that measures relative changes in ΔHbO and HbR in the brain. Its simplicity, affordability, and greater temporal resolution compared to fMRI make it widely accessible, allowing for multiple assessments on a subject. Importantly, fNIRS can be used during motor tasks to monitor motor network function and recovery in real time, making it particularly suitable for guiding individualized rehabilitation programs. Since ICH introduces rapid, hemisphere-specific reorganization in the acute phase (48–72 h), rehabilitation should be tailored to these early patterns. This opens the door for recovery strategies that go beyond physiotherapy, such as using a robotic-assisted gait and arm therapy to provide high-intensity training of affected movements. In addition, fNIRS-guided feedback can be incorporated to adapt therapy in real time, ensuring that training reinforces the networks showing the strongest compensatory potential. By aligning rehabilitation with each patient’s unique reorganization profile, clinicians can either strengthen the unaffected hemisphere’s preMC when restoration is less feasible, as in right-hemispheric ICH, or promote re-engagement of the affected hemisphere’s preMC and somatosensory cortices when early neuronal activity is observed, as in left-hemispheric ICH. Such personalized approaches, combining neuroimaging feedback, neuromodulation, and task-specific training, can have the potential to accelerate recovery in acute ICH.

## 5. Limitations

We acknowledge several important limitations of this study. First, the single-time-point design restricts our ability to capture definitive evidence of dynamic processes such as functional dysconnectivity or compensatory responses during recovery. Although our cross-sectional approach provides valuable early insights, a longitudinal follow-up is necessary to establish recovery trajectories and causal inferences. Our findings should be interpreted as exploratory and preliminary markers for such processes. Second, the placement of optodes was limited to motor, premotor, and somatosensory cortices, which constrained our ability to assess reorganization across the whole cortex; future work incorporating broader coverage will provide a more comprehensive view. Third, while our cohort of 35 patients and 39 controls is substantial for acute ICH research, subgrouping by hemisphere reduced the effective sample size, limiting statistical power and the ability to perform subgroup analyses. This limitation underscores the need for larger, multi-center cohorts to improve generalizability and to account for heterogeneity in ICH presentation. Fourth, although we employed individual Satori maps to mitigate individual-level variability, we did not correlate connectivity or oxygenation changes with clinical outcomes such as NIHSS or mRS, which would strengthen the translational relevance of our findings; integrating such measures is a priority for future studies. Fifth, while increased coherence was interpreted as a compensatory mechanism, we recognize the possibility of maladaptive hyperconnectivity, which warrants careful longitudinal investigation. These constraints highlight the exploratory nature of our study but also lay a strong foundation for future research. Specifically, longitudinal, multi-site studies with larger and more diverse cohorts, expanded cortical optode coverage, integration of outcome measures, and advanced signal correction approaches will help refine our understanding of motor network reorganization and recovery after acute-ICH.

## 6. Conclusions

This study provides insights into hemisphere-based differences in motor network disruptions following ICH. Left hemispheric preMC may be involved in motor cortex reorganization in acute ICH. These results highlight the importance of tailored rehabilitation strategies, with a greater emphasis on physical and occupational therapy for left hemispheric ICH patients. The left preMC may be a target for neuromodulation devices. However, larger cohort studies with standardized recruitment criteria are needed to validate these findings and further refine rehabilitation approaches for stroke patients.

## Figures and Tables

**Figure 1 sensors-25-06377-f001:**
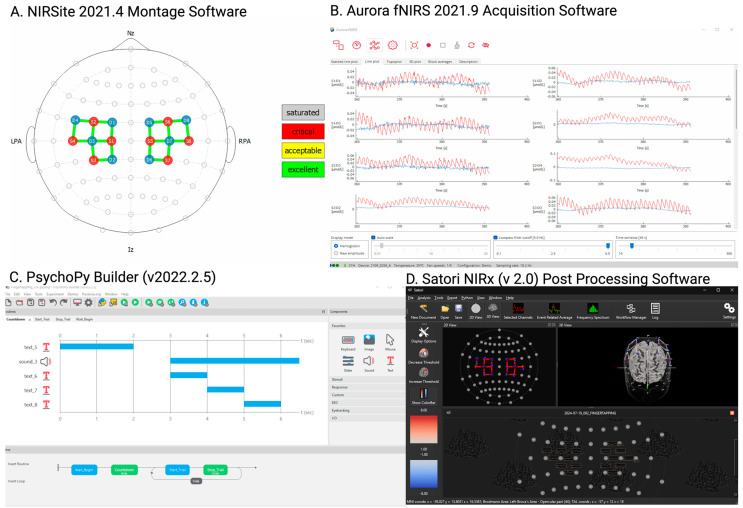
Experimental design overview: (**A**) motor cortex montage, (**B**) data acquisition software, (**C**) motor paradigm (Finger tapping) setup, and (**D**) post-processing software. (Created in BioRender. Kate, M., https://BioRender.com/8lddwmi, accessed on 2 October 2025).

**Figure 2 sensors-25-06377-f002:**
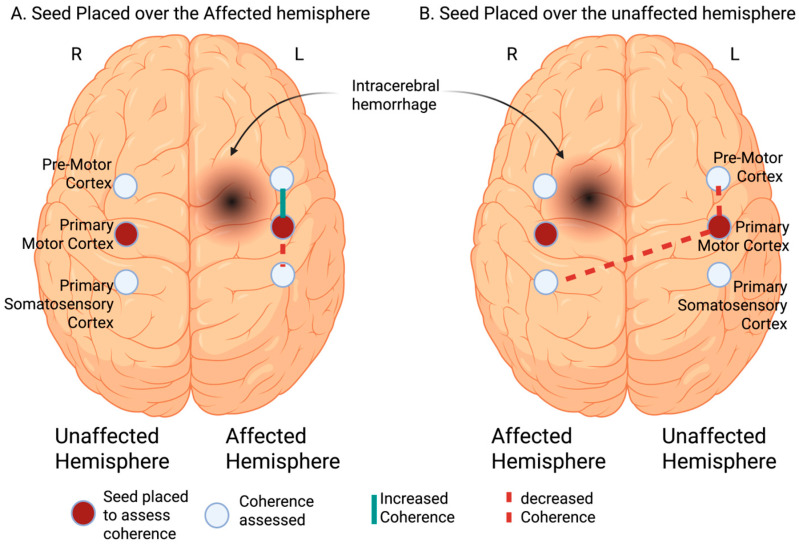
Resting state functional connectivity changes in patients with intracerebral hemorrhage compared to control participants. The seed was placed in the primary motor cortex (priMC) to assess the correlation with different cortical areas: (**A**) In patients with left hemispheric ICH and a seed placed over the left priMC cortex, increased correlation was noted with the affected premotor cortex and decreased correlation with the affected somatosensory cortex compared to the control participants. (**B**) In patients with right hemispheric ICH and a seed placed over the left hemisphere priMC, decreased correlation was observed with the affected somatosensory cortex and unaffected premotor cortex compared to control participants. (Created in BioRender. Kate, M. https://BioRender.com/0cnsq5x, accessed on 2 October 2025).

**Figure 3 sensors-25-06377-f003:**
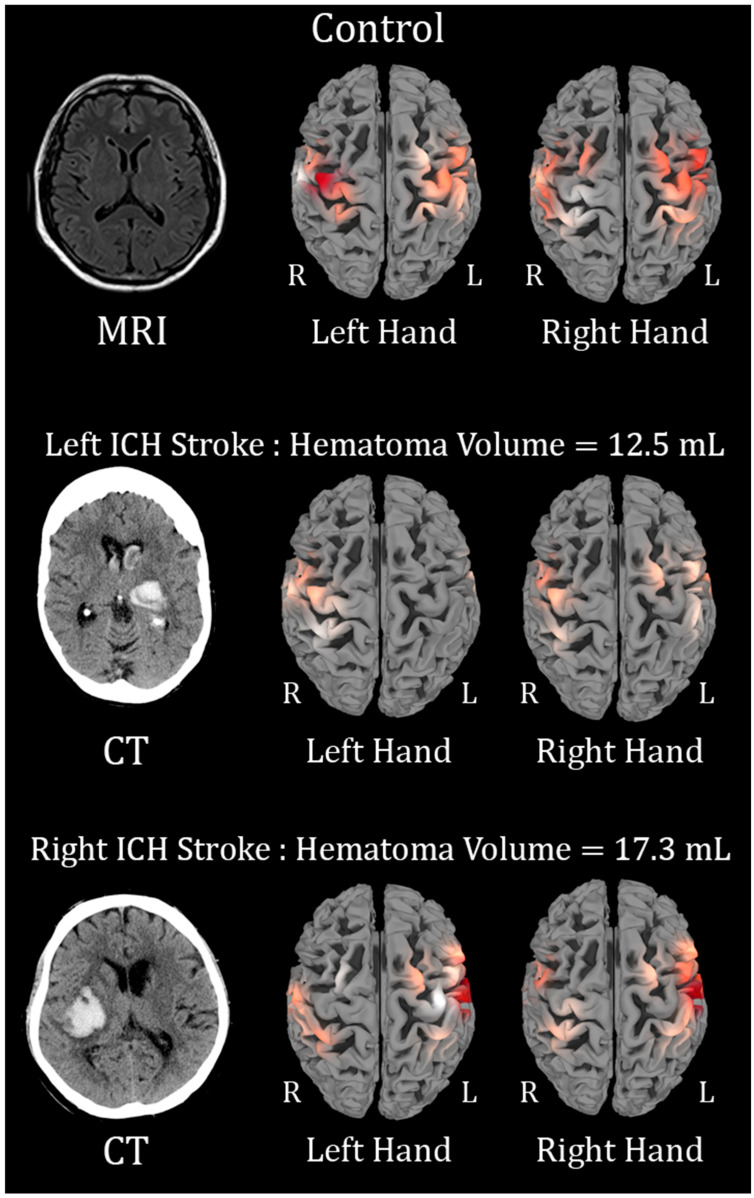
Brain activation maps showing ΔHbO changes during finger tapping (FT). Red regions indicate activation; R and L denote right and left hemispheres. In the (**first row**), the control subject, with left-hand FT, leads to activation of the R > L hemisphere, and right-hand FT leads to activation of the L > R. In the (**second row**), in a patient with a left hemisphere intracerebral hemorrhage (ICH), left-hand FT shows activation over the R only, and right-hand FT shows mild activation of the R & L activation, suggestive of motor cortex re-organization and spread of activation. In the (**third row**), in a patient with a right hemispheric ICH, the left-hand FT shows activation L > R (opposite compared to the control subject), and the right-hand FT shows activation L > R. This is suggestive of motor-cortex re-organization and spread of activation more to the L hemisphere. The color of red signals represents the concentration of ΔHbO; lighter to white signals represent low ΔhbO, and darker red signals represent high ΔHbO.

**Figure 4 sensors-25-06377-f004:**
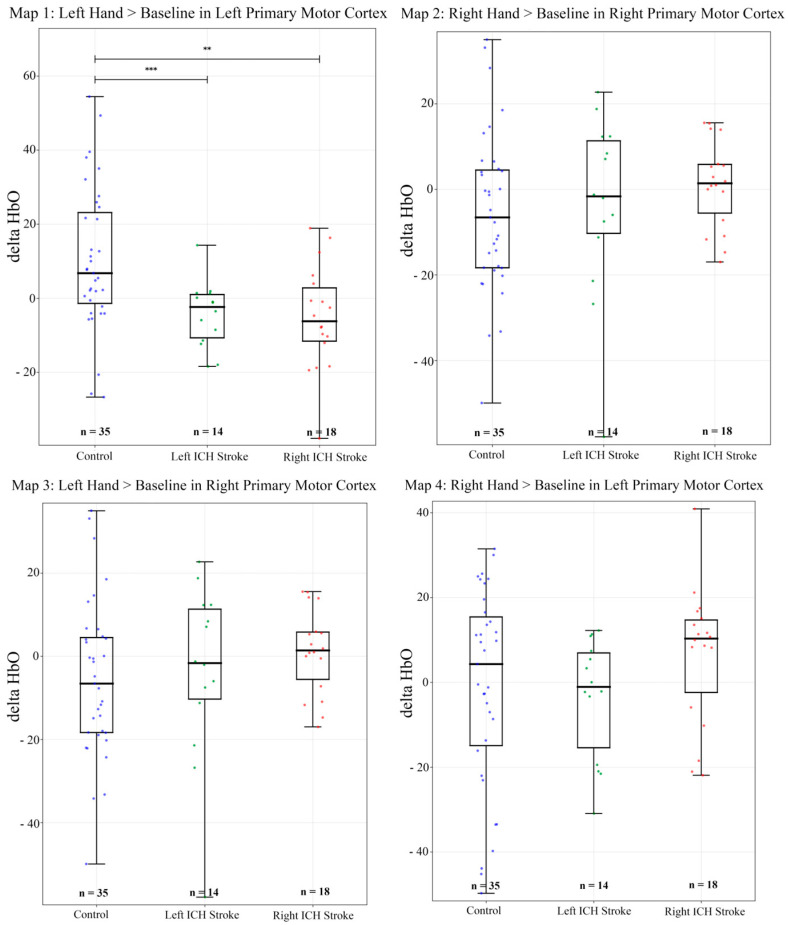
General linear model regression boxplots showing ΔHbO changes during left- and right-hand finger tapping across Control, Left/Right ICH Stroke. Statistical differences (** *p* < 0.05, *** *p* < 0.001) are indicated. Created in BioRender. Kate, M, https://BioRender.com/ddh0sgw, accessed on 2 October 2025.

**Table 1 sensors-25-06377-t001:** Baseline characteristics of the intracerebral hemorrhage (ICH) and control groups.

	Intracerebral Hemorrhage, *n* = 35	Control or TIA, *n* = 39	*p*-Value
Median (IQR) Age, Years	64 (56,83)	67 (58,75)	0.5
Female Sex, n (%)	15 (42.8)	17 (43.5)	0.9
Median Premorbid (IQR) mRS	0 (0,1)	0 (0,0)	0.3
Hypertension, n (%)	29 (82.9)	24 (64.1)	0.07
Diabetes mellitus, n (%)	5 (14.3)	11 (28.2)	0.1
Dyslipidemia, n (%)	19 (54.3)	30 (76.9)	0.040
Atrial Fibrillation, n (%)	6 (17.1)	5 (12.8)	0.602
Coronary Artery Disease, n (%)	2 (5.9)	3 (7.7)	0.735
Small Vessel Disease, n (%)	24 (68.6)	5 (12.8)	0.000
Median (IQR) NIHSS at enrollment	10 (5,17)	-	-
Median (IQR) ICH Volume, mL	18.7 (7.2, 51.7)	-	-
Hand Weakness Present, n (%)	26 (74.3)	-	-
Location of ICH: Lobar, n (%) Non-Lobar, n (%) Both, n (%)	9 (25.7) 22 (62.9) 4 (11.4)	-	-
Median NIHSS at Discharge	5.5 (2.5, 10)	-	-
Median Hospital Stay Days	28 (4, 32)	-	-

TIA, transient ischemic attack; IQR, interquartile range; NIHSS, National Institute of Health Stroke Scale.

## Data Availability

Data will be available upon request to the corresponding author.

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
