# Peer review of "Early Motor Cortex Connectivity and Neuronal Reactivity in Intracerebral Hemorrhage: A Continuous-Wave Functional Near-Infrared Spectroscopy Study"

_sensors, 2025, doi:10.3390/s25206377_

Round 1
Reviewer 1 Report
Comments and Suggestions for Authors
Manuscript: sensors-3864092
Title: Early Motor Cortex Dysconnectivity and Compensatory Neuronal Reactivity in Intracerebral hemorrhage: a Continuous-wave functional Near-infrared Spectroscopy Study
This manuscript presents a valuable investigation into the early neurophysiological changes following ICH using continuous-wave functional near-infrared spectroscopy. The study's strength lies in its timely assessment of patients within 48-72 hours of onset, a crucial and often under-explored period for understanding brain reorganization.
The findings are compelling and contribute meaningfully to the field. The observation that resting-state functional connectivity (RSFC) and motor-task-related hemodynamic changes differ based on the stroke's laterality is particularly noteworthy. The specific finding of increased RSFC between the affected priMC and preMC in left ICH and decreased RSFC in right ICH provides novel insights into hemispheric-specific compensatory mechanisms.
Questions
- How can you infer a dynamic process of "functional disconnections" and "compensatory responses" from a static snapshot? Have you followed up with these patients to assess their long-term functional recovery?
- Your fNIRS probes were placed exclusively over the motor, premotor, and somatosensory cortices. By focusing on only three regions, aren't your conclusions about the overall reorganization pattern potentially biased and incomplete?
- How can you be sure that GSR did not eliminate genuine, ICH-related, whole-brain synchronized changes? Did you perform a sensitivity analysis to evaluate the impact of GSR on your results?
- Do your results have sufficient statistical power? Did you correct for multiple comparisons, such as using Bonferroni or False Discovery Rate (FDR) correction, given that you analyzed connectivity across multiple channels?
- You highlight the importance of the left preMC as a potential target for neuromodulation. However, you provide no evidence that your fNIRS findings directly correlate with patient clinical outcomes, such as their modified Rankin Scale (mRS) scores. If these hemodynamic changes do not predict a patient's recovery, what is their clinical relevance?
- Given the heterogeneity of ICH (e.g., location of bleeding, hematoma size, age), how can you be sure that these findings are generalizable to a broader patient population? Did you conduct any subgroup analyses based on these variables?
Author Response
Comment 1: How can you infer a dynamic process of "functional disconnections" and "compensatory responses" from a static snapshot? Have you followed up with these patients to assess their long-term functional recovery?
Response 1: We acknowledge that a key limitation of our study is the single-time-point assessment, which restricts our ability to directly infer dynamic processes, such as functional disconnections or compensatory responses, over time. While longitudinal follow-up was not included in this study, it is a planned focus for future studies. Our use of both resting-state and motor paradigms provides a snapshot of functional organization, and comparisons with control subjects further strengthen the validity of our findings. These cross-sectional insights offer meaningful evidence that warrants longitudinal investigation. We have changed the title to the following: “Early motor cortex connectivity and neuronal reactivity in Intracerebral hemorrhage: a continuous-wave functional near infrared spectroscopy study.” We updated discussions 438-445 and our limitations section 505 to 536.
Comment 2: Your fNIRS probes were placed exclusively over the motor, premotor, and somatosensory cortices. By focusing on only three regions, aren't your conclusions about the overall reorganization pattern potentially biased and incomplete?
Response 2: We agree with the reviewer that we exclusively placed the source and detectors over the motor, premotor, and somatosensory cortices, which restricts our ability to assess reorganization across the entire cortex. Weakness due to affection of the motor system circuit (pyramidal tract) is one of the most common symptoms after intracerebral hemorrhage. Weakness can be easily and objectively assessed with the National Institute of Health Stroke Scale (NIHSS). Thus, the motor cortex and its surrounding areas were assessed in this study. While our conclusions are specific to these areas, they align with prior functional Magnetic Resonance Imaging studies (fMRI) that have demonstrated motor cortex reorganization in acute ICH. We acknowledge that broader cortical coverage would provide a more comprehensive view and plan to incorporate this in future studies. We updated our limitations section 505 to 536.
Comment 3: How can you be sure that GSR did not eliminate genuine, ICH-related, whole-brain synchronized changes? Did you perform a sensitivity analysis to evaluate the impact of GSR on your results?
Response 3:
We appreciate the reviewer’s concern regarding Global Signal Regression (GSR), implemented in our pipeline as Global Component Regression (GCR). To minimize the risk of removing genuine ICH-related whole-brain synchronized activity, we set the GCR component to its lowest setting (1), targeting only physiological noise sources such as respiration and cardiac signals. While we did not perform a formal sensitivity analysis, we did include Temporal Derivative Distribution Repair (TDDR) and Monotonic Interpolation in our spike removal parameter, both of which are intended to smooth out abrupt signals. These steps ensured that a global signal was derived from clean data, minimizing the risk of GCR eliminating genuine ICH-related synchronized activity. Both steps were applied prior to GCR in our pre-processing pipeline. We have described this on page 7, lines 224-230.
Comment 4: Do your results have sufficient statistical power? Did you correct for multiple comparisons, such as using Bonferroni or False Discovery Rate (FDR) correction, given that you analyzed connectivity across multiple channels?
Response 4: We thank the reviewer for raising an important point.
Our study results are exploratory and hypothesis-generating, and need further validation from other groups. We have added this to our limitation section in lines 534-539.
Our statistical analysis incorporated False Discovery Rate (FDR) correction during the application of GLM and across all contrast comparisons to account for multiple comparisons, and we have updated our methods section, lines 298 to 302.
Comment 5: You highlight the importance of the left preMC as a potential target for neuromodulation. However, you provide no evidence that your fNIRS findings directly correlate with patient clinical outcomes, such as their modified Rankin Scale (mRS) scores. If these hemodynamic changes do not predict a patient's recovery, what is their clinical relevance?
Response 5: We agree with the reviewer. We had 19 (54.3%) patients with right hemispheric ICH. We have assessed the modified Rankin scale at 90 days, with a median of 4. We performed these analyses. We have edited sections 299-303 of the statistical analyses and sections 422-427 of the results.
Comment 6: Given the heterogeneity of ICH (e.g., location of bleeding, hematoma size, age), how can you be sure that these findings are generalizable to a broader patient population? Did you conduct any subgroup analyses based on these variables?
Response 6: We thank the reviewer for highlighting the issue of heterogeneity in ICH, which is a factor to consider when evaluating generalizability. Due to the limited sample size in our study, we did not conduct a formal subgroup analysis. However, to account for individual variability, we generated individual-based Satori maps, which showcased neuronal activity across motor cortices, allowing us to visualize and compare activation patterns on a per-patient basis (this will be uploaded as supplementary data). We acknowledge that without subgroup analysis, broader generalization remains limited. Further studies can include larger and more diverse samples, which will be essential to explore how specific clinical variables influence cortical reorganization and generalize findings to a wider ICH population. We updated our limitations section 505 to 536.
Reviewer 2 Report
Comments and Suggestions for Authors
The manuscript used continuous wave functional near-infrared spectroscopy (cw-fNIRS) technology to evaluate early motor cortex functional connectivity (resting state functional connectivity, RSFC) and neural reactivity changes in patients with acute intracerebral hemorrhage (acute ICH). The study found that in patients with left-sided brain hemorrhage, the RSFC between the affected primary motor cortex (priMC) and the affected premotor cortex (preMC) increased; whereas in patients with right-sided brain hemorrhage, the RSFC between the unaffected priMC and the affected sensory cortex decreased. Additionally, during motor tasks, left-sided brain hemorrhage patients showed an increase in oxygenated hemoglobin (ΔHbO) in the affected preMC and sensory cortex when performing left finger tapping; right-sided brain hemorrhage patients showed an increase in ΔHbO in the unaffected preMC during left finger tapping. These findings suggest that motor cortex reorganization in acute ICH exhibits laterality differences, and the left preMC may become a target for neuroregulatory interventions.
However, there are several aspects that need improvement:
Comments:
- It is recommended to improve the clarity of Figure 1 to ensure that all labels and chart elements are clear and readable.
- It is recommended to increase the size of data points in Figure 3 to improve the aesthetic and readability of the data analysis graph.
- It is recommended to provide both the full form and abbreviation of terms when they first appear in the text, and then use the abbreviation in subsequent mentions.
- The phrase "4, -0.13, p = 0.02), compared to control/TIA" on line 330 should be moved to line 315.
- It is recommended to change the statistical symbols "β" and "p" to italics throughout the text to comply with scientific publication standards.
- It is recommended to further discuss how these findings can be translated into specific rehabilitation measures in the discussion section.
- It is recommended to conduct longitudinal analysis in future studies to assess how RSFC and neural reactivity change over time.
This study reveals the early changes in motor cortex functional connectivity and neural reactivity after ICH through cw-fNIRS technology and identifies laterality differences. These findings provide new evidence for the development of personalized rehabilitation strategies. It is suggested that the authors address the above issues to improve the quality and clinical applicability of the paper.
Author Response
Comment 1: It is recommended to improve the clarity of Figure 1 to ensure that all labels and chart elements are clear and readable.
Response 1: We appreciate the reviewer's helpful suggestion. In response, we have enhanced the clarity of Figure 1 by improving label resolution, adjusting font sizes, and refining chart elements to ensure that all components are clearly visible and easily interpretable.
Comment 2: It is recommended to increase the size of data points in Figure 3 to improve the aesthetic and readability of the data analysis graph.
Response 2: We thank the reviewer for the helpful suggestion. In response, we have increased the size of the data points in Figure 3 to enhance both the aesthetic quality and readability of the box plot.
Comment 3: It is recommended to provide both the full form and the abbreviation of terms when they first appear in the text, and then use the abbreviation in subsequent mentions.
Response 3: We thank the reviewer for this recommendation. In response, we have revised the manuscript to ensure that all technical terms are introduced with both their full form and corresponding abbreviation upon first mention. Subsequent references consistently use the abbreviation to maintain clarity and readability throughout the text.
Comment 4: The phrase "4, -0.13, p = 0.02), compared to control/TIA" on line 330 should be moved to line 315.
Response 4: We thank the reviewer for the careful observation. In response, we have relocated the phrase "4, -0.13, p = 0.02), compared to control/TIA" from line 330 to line 325 in the revised manuscript, to improve clarity and contextual alignment within the paragraph.
Comment 5: It is recommended to change the statistical symbols "β" and "p" to italics throughout the text to comply with scientific publication standards.
Response 5: We thank the reviewer for this valuable recommendation. In response, we have updated the manuscript to ensure that all statistical symbols, including β and p, are presented in italics throughout the text in accordance with scientific publication standards.
Comment 6: It is recommended to further discuss how these findings can be translated into specific rehabilitation measures in the discussion section.
Response 6: We thank the reviewer for this insightful recommendation. In response, we have expanded the discussion section on line 487 to explore how our findings may inform targeted rehabilitation strategies based on stroke laterality. Specifically, we discuss how differential activation patterns in the left/right hemispheric strokes could guide individualized neuromodulation and motor training rehabilitation.
Comment 7: It is recommended to conduct longitudinal analysis in future studies to assess how RSFC and neural reactivity change over time.
Response 7: We agree that conducting longitudinal analyses would provide critical insight into how RSFC and neural reactivity evolve over time, particularly in the context of post-ICH recovery. While our current study was limited to a single time point, we recognize the importance of capturing dynamic changes in cortical activity to better understand the trajectory of rehabilitation potential. We have expanded the limitations section on this.
Reviewer 3 Report
Comments and Suggestions for Authors
I have the following comments to improve the article:
-
The abstract should briefly mention the number of participants excluded due to poor signal quality to clarify the final analyzed sample size early in the paper.
-
In the methods, provide more detail about how motion artifacts were handled (e.g., specify the algorithm parameters for spike removal and TDDR) so that readers can replicate the study.
-
The statistical analysis section should specify whether multiple comparison corrections (e.g., Bonferroni, FDR) were applied during RSFC and GLM analyses to control for Type I errors.
- Include effect sizes (e.g., Cohen’s d or partial eta-squared) alongside p-values in the results to help readers interpret the strength of effects.
-
Add a visual summary (schematic diagram) of the main RSFC findings (increased vs. decreased connectivity for left vs. right ICH) to improve clarity.
-
Expand the discussion to include practical implications for rehabilitation planning and potential neuromodulation targets.
-
Strengthen the limitations section by explicitly stating that the single time-point design limits causal inference and proposing longitudinal follow-up studies.
-
Fix formatting error in line 296: "42:1 (22:6, 88:3)" likely should be "42.1 (22.6, 88.3)" or similar.
-
Maintain consistent terminology and capitalization for abbreviations such as preMC, priMC, RSFC throughout the manuscript.
Author Response
Comment 1: The abstract should briefly mention the number of participants excluded due to poor signal quality to clarify the final analyzed sample size early in the paper.
Response 1: We thank the reviewer for this suggestion. We have updated lines 38 and 40 of the abstract to mention the final analyzed subjects, excluding those participants with poor signal quality.
Comment 2: In the methods, provide more detail about how motion artifacts were handled (e.g., specify the algorithm parameters for spike removal and TDDR) so that readers can replicate the study.
Response 2: We thank the reviewer for this important recommendation. In response, we have included the spike removal parameters in the supplementary section and provided figures of our pre-processing pipelines (Supplementary Figure 1-3), which include all the parameters used for processing the raw data, along with their settings. These additions aimed to enhance transparency and ensure replicability by other researchers.
Comment 3: The statistical analysis section should specify whether multiple comparison corrections (e.g., Bonferroni, FDR) were applied during RSFC and GLM analyses to control for Type I errors.
Response 3: We thank the reviewer for this observation. In response, we have updated the Statistical Analysis section to specify that multiple comparison corrections were applied during both RSFC and GLM analyses to control for Type I errors. Specifically, we used the False Discovery Rate (FDR) correction method to adjust for multiple comparisons across channels and conditions on lines 271 to 275.
Comment 4: Include effect sizes (e.g., Cohen’s d or partial eta-squared) alongside p-values in the results to help readers interpret the strength of effects.
Response 4: We thank the reviewer for this suggestion. In response, we have updated the Results section to include effect sizes, specifically Cohen’s d, alongside p-values and effect sizes for all relevant group comparisons with respect to Figure 3. The changes can be found from lines 380 to 400.
Comment 5: Add a visual summary (schematic diagram) of the main RSFC findings (increased vs. decreased connectivity for left vs. right ICH) to improve clarity.
Response 5: We have added Figure 3, which describes the RSFC findings.
Comment 6: Expand the discussion to include practical implications for rehabilitation planning and potential neuromodulation targets.
Response 6: We thank the reviewer for this important recommendation. In response, we have expanded the discussion section to include the practical implications of our findings for rehabilitation planning on line 487.
Comment 7: Strengthen the limitations section by explicitly stating that the single time-point design limits causal inference and proposing longitudinal follow-up studies.
Response 7: We thank the reviewer for this recommendation. In response, we have strengthened the limitations section on line 504 by explicitly acknowledging that the single time-point design of our study restricts our ability to draw causal inferences about the observed changes in RSFC and neural reactivity. To address this limitation, we have proposed longitudinal follow-up studies that would allow for tracking recovery over time and better understanding the trajectory of rehabilitation potential in post-ICH patients.
Comment 8: Fix formatting error in line 296: "42:1 (22:6, 88:3)" likely should be "42.1 (22.6, 88.3)" or similar.
Response 8: We thank the reviewer for pointing out the formatting error. In response, we have corrected the values on line 296 to read "42.1 (22.6, 88.3)" to reflect the median and IQR using standard decimal notation accurately.
Comment 9: Maintain consistent terminology and capitalization for abbreviations such as preMC, priMC, RSFC throughout the manuscript.
Response 9: We thank the reviewer for this observation. In response, we have carefully reviewed the manuscript to ensure consistent use of terminology and capitalization for all abbreviations, including preMC, priMC, and RSFC.
Reviewer 4 Report
Comments and Suggestions for Authors
In this manuscript "Early Motor Cortex Dysconnectivity and Compensatory Neuronal Reactivity in Intracerebral Hemorrhage: a Continuous-wave functional Near-infrared Spectroscopy Study", The authors explore early motor cortex network changes in patients with intracerebral hemorrhage using functional near-infrared spectroscopy. The integration of resting-state functional connectivity with task-based paradigms makes the work stand out, as it offers a dual perspective on cortical reorganization. The manuscript is well organized, but certain areas require clarification and deeper reflection before it can reach its full impact.
Major concern
-
Sample size and subgrouping
The final analysis includes 35 patients and 39 controls. While this is a respectable cohort given the challenges of acute ICH studies, once divided into left- and right-hemisphere subgroups the effective sample size becomes modest. It would be important to explain whether power calculations were performed or at least acknowledge the limited statistical power when interpreting negative results. -
Statistical analysis
Multiple channels and comparisons were tested, but I could not find mention of any correction for multiple comparisons (e.g., Bonferroni, FDR). Without such correction, some of the reported significant effects may represent chance findings. Even if formal correction is not applied, the authors should explicitly state this limitation and interpret marginal results cautiously. -
Cross-sectional design
The study examines patients at a single time point, yet some parts of the discussion suggest longitudinal inferences (e.g., trajectories of reorganization). These statements should be toned down or rephrased. Alternatively, the authors could highlight how their findings provide a rationale for longitudinal follow-up in future work. -
Clinical relevance
The data are compelling, but the translational angle would be strengthened by linking the connectivity/oxygenation findings to clinical measures such as NIHSS or mRS. Even exploratory correlations would provide valuable context and demonstrate how the neurophysiological findings relate to patient outcomes. -
Interpretation of connectivity changes
Increased coherence is consistently interpreted as compensatory. While this is plausible, the literature also recognizes maladaptive hyperconnectivity. The authors should discuss both possibilities and explain why they favor the compensatory explanation in this context.Minor Points
-
The introduction is dense, with long sentences and heavy use of abbreviations. Shortening and reintroducing key terms more frequently will make the paper easier for non-specialists to follow.
-
Figures are useful but the legends could be expanded to clearly explain what the color scales and statistical markers represent.
-
Table 1 could be reformatted for easier comparison of comorbidities.
-
The limitations section is appropriately candid, but the issue of superficial signal contamination (absence of short channels) deserves a clearer explanation of how it might have influenced the results.
-
Several references have inconsistent formatting; careful editing will help the manuscript read more smoothly.
Recommendation
The study is innovative and has real potential to influence the field, but the issues outlined above particularly statistical handling, interpretative balance, and clearer clinical linkage should be addressed.
Once revised, the work would make a valuable contribution to the literature on acute stroke recovery and the application of fNIRS in clinical neuroscience
-
Author Response
Comment 1: Sample size and subgrouping
The final analysis includes 35 patients and 39 controls. While this is a respectable cohort given the challenges of acute ICH studies, once divided into left- and right-hemisphere subgroups, the effective sample size becomes modest. It would be important to explain whether power calculations were performed or, at the very least, acknowledge the limited statistical power when interpreting negative results.
Response 1: We thank the reviewer for this thoughtful comment. This study was designed as an exploratory investigation into RSFC and cortical activation patterns following acute ICH. Formal power calculations were not performed due to the exploratory nature of the study. We have now expanded the limitations section to explicitly note that the reduced subgroup sizes may constrain our ability to detect subtle effects. We also emphasized that future studies with larger, longitudinal cohorts are needed to validate and extend these preliminary findings.
Comment 2: Statistical analysis
Multiple channels and comparisons were tested, but I could not find any mention of a correction for multiple comparisons (e.g., Bonferroni, false discovery rate, FDR). Without such correction, some of the reported significant effects may represent chance findings. Even if formal correction is not applied, the authors should explicitly state this limitation and interpret marginal results cautiously.
Response 2: We thank the reviewer for highlighting this important consideration. In response, we have clarified in the Methods section on lines 271 to 275 that FDR correction was applied to account for multiple comparisons across channels.
Comment 3: Cross-sectional design
The study examines patients at a single time point, yet some parts of the discussion suggest longitudinal inferences (e.g., trajectories of reorganization). These statements should be toned down or rephrased. Alternatively, the authors could highlight how their findings provide a rationale for longitudinal follow-up in future work.
Response 3: We thank the reviewer for this important point. We acknowledge that a single-point cross-sectional design limits our ability to draw definitive conclusions about neural reorganization and compensation. In response, we have discussed this further in the limitations section to clarify that our findings represent preliminary evidence of compensatory changes in motor cortices and activation patterns following acute ICH. We have edited the highlights (lines 22-27, lines 31), the conclusion in the abstract (line 61) and the manuscript (line 592).
Comment 4: Clinical relevance
The data are compelling, but the translational angle would be strengthened by linking the connectivity/oxygenation findings to clinical measures such as NIHSS or mRS. Even exploratory correlations would provide valuable context and demonstrate how the neurophysiological findings relate to patient outcomes.
Response 4: We appreciate the reviewer’s important suggestion. We performed these analyses. We have edited sections 299-303 of the statistical analyses and sections 422-427 of the results.
Comment 5: Interpretation of connectivity changes
Increased coherence is consistently interpreted as a compensatory mechanism. While this is plausible, the literature also recognizes maladaptive hyperconnectivity. The authors should discuss both possibilities and explain why they favor the compensatory explanation in this context.
Response 5: We appreciate the reviewer’s important idea. In response, we have expanded the discussion on lines 438 to 446 to acknowledge that increased coherence may reflect either compensatory reorganization or maladaptive hyperconnectivity or both, as recognized in the broader neuroimaging literature.
Comment 6: The introduction is dense, with long sentences and heavy use of abbreviations. Shortening and reintroducing key terms more frequently will make the paper easier for non-specialists to follow.
Response 6: We thank the reviewer for this helpful observation. In response, we have revised the Introduction to improve readability by shortening complex sentences, reducing the density of abbreviations, and reintroducing key terms more frequently.
Comment 7: Figures are useful but the legends could be expanded to clearly explain what the color scales and statistical markers represent.
Response 7: We appreciate the reviewer’s feedback regarding the figure legends. In response, we have expanded Figure 2's legend by providing a clear explanation of what the colour scales mean in Figure 2's caption on line 349.
Comment 8: Table 1 could be reformatted for easier comparison of comorbidities.
Response 8: Since the control subjects do not have the baseline neurological deficits and lack the intracerebral hemorrhage volumetrics, Table 1 appears asymmetric.
Comment 9: The limitations section is appropriately candid, but the issue of superficial signal contamination (absence of short channels) deserves a clearer explanation of how it might have influenced the results.
Response 9: We thank the reviewer for this point. We have appropriately revised the limitations section and brought the necessary points into consideration. Regarding short channels, after reviewing our pre-processing pipeline, we have acquired and applied short channel regression, thereby eliminating this limitation.
Comment 10: Several references have inconsistent formatting; careful editing will help the manuscript read more smoothly.
Response 10: We have now organized our references as per journal requirements.
Round 2
Reviewer 2 Report
Comments and Suggestions for Authors
The manuscript has been revised and supplemented in accordance with the comments raised, and in my opinion, it now meets the publication standards of this journal.
Reviewer 3 Report
Comments and Suggestions for Authors
Paper can be accepted now. However, please revise the English.